# Prevalence and Associated Factors of E-Cigarette Use among Adolescents in Southeast Asia: A Systematic Review

**DOI:** 10.3390/ijerph20053883

**Published:** 2023-02-22

**Authors:** Miaw Yn Jane Ling, Ahmad Farid Nazmi Abdul Halim, Dzulfitree Ahmad, Norfazilah Ahmad, Nazarudin Safian, Azmawati Mohammed Nawi

**Affiliations:** Department of Community Health, Faculty of Medicine, Universiti Kebangsaan Malaysia, Kuala Lumpur 56000, Malaysia

**Keywords:** adolescent, youth, e-cigarette, vape, Southeast Asia

## Abstract

The use of e-cigarettes in adolescents remains a major public health concern. Like other tobacco products, e-cigarettes pose health risks to adolescents. Understanding the magnitude of this problem and identification of its associated factors will serve as a guide for development of preventive interventions. This systematic review aims to identify and discuss current epidemiological data on the prevalence and associated factors of e-cigarette use among adolescents in Southeast Asia. The reporting of this systematic review is in accordance with the Preferred Reporting Items for Systematic Reviews and Meta-Analyses (PRISMA) 2020 statement. We carried out a literature search through three databases (Scopus, PubMed, Web of Science) and targeted original English-language articles published between 2012 and 2021. A total of 10 studies were included in this review. The prevalence of current e-cigarette uses ranges from 3.3% to 11.8%. Several associated factors of e-cigarette use were identified, including sociodemographic factors, traumatic childhood experience, peer and parental influence, knowledge and perception, substance use, and accessibility of e-cigarettes. These factors should be addressed though multifaceted interventions which simultaneously target multiple factors. Laws, policies, programs, and interventions must be strengthened and tailored to the needs of adolescents at risk of using e-cigarettes.

## 1. Introduction

E-cigarettes are becoming increasingly popular around the world [1,2]. They are devices which use an electrically powered coil to heat e-liquid solution to produce an aerosol that is inhaled [3]. Although the original devices resembled cigarettes, many of the current models resemble tech gadgets, pens, and other items (Figure 1) [4]. E-liquids often contain flavors and additives which are dissolved in a propylene glycol or glycerin solution, the principal carriers used, and usually, but not always, contain nicotine. Hence, there are two different types of e-cigarettes in use other than e-cigars and e-pipes, which are known as electronic nicotine delivery systems (ENDS) and sometimes electronic non-nicotine delivery systems (ENNDS) [5]. Nevertheless, nicotine and other dangerous compounds which are commonly found in e-cigarette emissions are harmful to both users and non-users who are exposed to the aerosols secondhand.

Adolescence is a vital period of development as it is during this time that many lifelong health habits are developed. This is a critical stage in the development of healthy behaviors such as not smoking, and habits formed at this age are frequently carried over into adulthood [6,7,8]. The landscape of adolescent tobacco use is rapidly changing, with recent declines in combustible cigarette smoking and overall increases in the use of e-cigarettes and other nicotine delivery devices such as shisha [9,10,11]. The proportion of young people aged 13–15 years who use ENDS and ENNDS on a regular basis is higher than among their adult counterparts according to data among young people aged 13–15 years from 22 countries [12]. Hence, preventing such habits at this period may be crucial to lowering the risk of several preventable chronic diseases later in life.

E-cigarettes first entered the United States market in mid-2000s and the sales have increased rapidly since 2007. The current e-cigarette use among adolescents in the United States has increased by 900% between 2011 and 2015 [13]. Between 2017 and 2019, figures for young individuals ranged from 0.7% in Japan to 18.4% in Ukraine, with a median country value of 8.1% [12]. Between 2008 and 2015, the number of young people who had ever used ENDS and ENNDS increased in Poland, New Zealand, South Korea, and the United States, decreased in Canada and Italy, and remained consistent in the United Kingdom [14].

Unlike in high income countries, several countries in Southeast Asia observed increasing use of e-cigarettes later (since 2015) [15,16,17]. Reports indicated that the prevalence of e-cigarette use among Malaysian adolescents increased by more than 700% from 1.2% to 9.8% [16,17]. According to the Global Youth Tobacco Survey, 3.3% of adolescents currently use e-cigarettes in Thailand [18], while 14.1% of adolescents currently use e-cigarettes in Philippines [19].

E-cigarette sales in six Southeast Asian countries totaled $595 million in 2019, with an expected increase to $766 million by 2023 [20]. Currently, the Southeast Asia region is being targeted by the e-cigarette industry as they have a large smoking population with a growing e-cigarette market. Youths in Southeast Asia are targeted by a broad range of flavors, trendy designs, and point of sale promotion. Nevertheless, country policy responses in Southeast Asia vary considerably, from strict bans to no or partial regulation. In countries with weak e-cigarette regulations, the usage may be pervasive, especially among young people [20].

Even though Southeast Asia represents a potential area of future growth in the e-cigarette industry, e-cigarette research in this region has received relatively little attention. Previous systematic reviews focused on the association between e-cigarette use and subsequent cigarette smoking initiation among adolescents [21,22]. One recently published systematic review examined the international prevalence and associated factors of e-cigarette use among young people. However, the review is not extensive as it included mostly developed countries and only two included studies were conducted in Southeast Asia [23].

Determining the prevalence of e-cigarette use will assist in understanding the trend of this issue, while identification of factors associated with e-cigarette use will assist the development and implementation of interventions addressing the increasing trend of e-cigarette use among adolescents. Obtaining this information in the Southeast Asia region is important due to the differences in socioeconomic status, culture, and smoking norms. Hence, in this review, we aim to present and systematically analyze the prevalence and factors associated with e-cigarette use among adolescents in Southeast Asia together with future recommendations towards its prevention.

## 2. Materials and Methods

This systematic review is prepared in accordance with the PRISMA (Preferred Reporting Items for Systematic Reviews and Meta Analyses) updated guidelines [24]. The objective of this review is to determine the prevalence of e-cigarette use among adolescents in Southeast Asia and its associated factors. The component of mnemonic CoCoPop [25] (condition, context, population) and PEO [25] (population, exposure, outcome) were established as follows:Condition: Epidemiology of e-cigarette use;Context: Southeast Asia;Population: adolescents in Southeast Asia;Exposure: e-cigarette usage;Outcome: prevalence and associated factors.

### 2.1. Searching Strategy

The literature search was conducted in April 2022. For a comprehensive search, we used Web of Science, PubMed, and Scopus databases. The keywords used for searching of related articles are provided in Table 1. All the retrieved articles were imported into EndNoteX7 library, and library de-duplication was implemented according to Bramer et al. [26].

### 2.2. Eligibility Criteria

The inclusion criteria were: (1) publication in English language; (2) original articles which investigate the prevalence and associated factors for e-cigarette use among adolescents in Southeast Asia. In contrast, non-original articles such as conference proceedings, perspectives, commentaries, opinions, reports, systematic reviews, and meta-analyses were excluded. Since e-cigarettes were invented in 2003 [20], however, the publication period was limited from 2012 until 2021 to obtain the latest findings.

### 2.3. Study Selection

Five independent reviewers screened the title and abstract of the retrieved materials against the inclusive criteria. The potential articles identified during the main screening were kept, and the full text was independently reviewed by the same reviewers in detail according to the inclusion criteria. Disagreements were resolved through discussion and consensus among the five reviewers and input from a sixth reviewer.

A total of 51 articles were removed during the screening, leaving 11 articles for full-text screening. A total of 10 out of 11 articles were included, while one article published in 2022 was excluded (Figure 2).

### 2.4. Critical Appraisal and Data Extraction

For assessing the quality of analytical cross-sectional studies, the NIH quality assessment tool for observational cohort and cross-sectional studies, JBI critical appraisal checklist for analytical cross-sectional study, and the appraisal tool for cross-sectional studies are recommended tools. Among these three tools, the JBI checklist is the most preferred one [27]. Hence, the quality appraisal of this review was conducted by using the JBI critical appraisal checklist. 

Three reviewers extracted the data, which was then assessed independently by the other three reviewers. Table 2 and Table 3 show the results of the data extraction, which includes the authors, year of publication, title, study design, prevalence, and factors associated with e-cigarette use among adolescents in Southeast Asia.

## 3. Results

The search yielded 8 articles from SCOPUS and 2 articles from WOS, resulting in 10 unique hits. These articles were included in the full-text assessment after rigorous selection screening, as shown in the PRISMA flow diagram (Figure 2). A descriptive summary of the included studies in this review regarding study location and study design is presented in Table 2. The findings from 10 studies included in this systematic review are shown in Table 3. Four eligible articles were from Thailand and three each from Indonesia and Malaysia. The analyzed articles were published between 2017 and 2021. All articles included were cross-sectional studies.

### 3.1. Prevalence of E-Cigarette Use 

A total of nine articles mentioned about the prevalence of current e-cigarette smoking (past 30 days use) among adolescents in their respective research area, ranging from 3.3% from a study in Thailand by Chotbenjamaporn et al. [31] to 11.8% in a study in Indonesia by Bigwanto et al. [37]. Based on country, the prevalence of current e-cigarette smoking among adolescent is lowest in the studies conducted in Thailand, with range of 3.3% by Chotbenjamaporn et al. [31] to 6.7% by Ofuchi et al. and Thepthien et al. [28,30]. This is followed by studies conducted in Malaysia, which range from 5.9% by Yusof et al. [34] to 9.1% by Robert Lourdes et al. [32]. The highest prevalence of current e-cigarette smoking among adolescents in this study is in Indonesia, ranging from 10.7% by Kristina et al. [35] to 11.8% by Bigwanto et al. [37]. 

### 3.2. Associated Factors of E-Cigarette Use

#### 3.2.1. Sociodemographic (Gender, Age, Ethnicity, School Location, Academic Performance, and Sexual Experience)

Of 10 studies included, 7 mentioned male gender as risk factor for e-cigarette use among adolescents. One study mentioned age (older), ethnicity (Malays and Sabah and Sarawak Bumiputeras) and school location (urban) as another significant risk factor for e-cigarette use among adolescents. Additionally, one study identified associations between academic performance and sexual experience with e-cigarette use in adolescents. 

A study by Robert Lourdes et al. [32] shows that among Malaysian adolescents, Malay and Sabah and Sarawak Bumiputera adolescents have 2.25 higher odds of using e-cigarettes. A study in Indonesia by Fauzi and Areesantichai [36] also came to the conclusion that adolescent males in urban schools have higher risk of using e-cigarettes. In addition, a study by Robert Lourdes et al. [32] found that older adolescents have higher risks of e-cigarette use. Another recent study also found that those who had poor academic performance and had ever had sex were more likely to use e-cigarettes [28].

#### 3.2.2. Traumatic Childhood Experience

Only one study focuses on traumatic childhood experience as a risk factor for e-cigarette use among adolescents. A study in Thailand by Ofuchi et al. [30] describes how adverse childhood experience such as emotional abuse, physical abuse, sexual abuse, parental separation, child violence, and incarcerated households plays an important role in influencing the adolescents involved to start using e-cigarettes, with the prevalence of e-cigarette use of 6.7%.

#### 3.2.3. Peer and Parental Influence

Five articles discuss on the role of peer and parental influence as a risk factor for e-cigarette use among adolescents. A study by Patanavanich et al. [29] in Thailand shows that adolescents with parents or peers who are currently using e-cigarettes have four to six times higher odds of using e-cigarettes, while those who had peers who use cigarettes were also at risk of e-cigarette use [28]. Fauzi and Areesantichai mentioned that peer use was associated with two times higher odds of e-cigarette use [36]. Meanwhile, a study by Yusof et al. in Malaysia found that adolescents with peers who are e-cigarette users have 12 times higher odds of using e-cigarettes [34]. Bigwanto et al. report that adolescents with parents that accept e-cigarette use have almost four times higher odds of using e-cigarettes [37]. 

#### 3.2.4. Knowledge and Perception of E-Cigarettes

In this study, four articles describe the knowledge and perception of e-cigarettes among adolescents. Kristina et al. mentioned that adolescents in Indonesia have poor knowledge regarding the harmful effect of tobacco and e-cigarettes [35]. Patanavanich et al. found that adolescents in Thailand were unaware of the risk posed by e-cigarettes [29]. Another study in Indonesia by Fauzi and Areesantichai reports that adolescents have the perceptions that electronic cigarettes aid conventional cigarette smoking cessation, which is a risk factor itself for them to use e-cigarettes [36]. Meanwhile, Bigwanto et al. explained how adolescents in Indonesia have perception that e-cigarettes are less addictive than conventional cigarettes and e-cigarettes do not cause cancer [37].

#### 3.2.5. Substance Use (Cigarette Smoking and Alcohol Drinking)

In this study, seven articles mentioned the past history or current use of cigarettes as a risk factor for e-cigarette use among adolescents. A study in Indonesia by Kristina et al. found that adolescents with current usage of alcohol have three times higher odds of using e-cigarettes [35].

#### 3.2.6. Accessibility of E-Cigarette

Two articles discuss the role of accessibility of e-cigarettes as risk factors for e-cigarette use in adolescents. Fauzi and Areesantichai mentioned that easy access to e-cigarettes is among the risk factors for e-cigarette use in adolescents [36]. Meanwhile, having enough money to buy e-cigarettes is also positively associated with e-cigarette use among adolescents [37].

### 3.3. Quality Appraisal

Assessment of the included studies using the JBI critical appraisal checklist indicated that five studies fulfilled all nine criteria. The details of this assessment are reported in Table 4. 

## 4. Discussion

E-cigarette use among adolescents remains a serious public health concern. Most e-cigarettes contain nicotine, which is highly addictive and can harm the developing adolescent brain [13]. The adverse health effects of e-cigarette use, including pulmonary dysfunction and oxidative stress, have also been demonstrated in a previous study [38]. 

Monitoring tobacco use, including the use of e-cigarettes in adolescents, is an essential activity to combat the global tobacco epidemic as stipulated in the World Health Organization (WHO) Framework Convention on Tobacco Control (FCTC) MPOWER strategies [12]. Understanding the factors associated with e-cigarette use in adolescents will serve as a guide to support development and implementation of targeted interventions. In this systematic review of 10 studies, we summarized the prevalence of e-cigarette use among adolescents in Southeast Asia. We also identified several associated factors of e-cigarette use, which are classified into six categories: (1) sociodemographic, (2) traumatic childhood experience, (3) peer and parental influence, (4) knowledge and perception of e-cigarettes, (5) substance use, and (6) accessibility of e-cigarettes.

### 4.1. Prevalence of E-Cigarette Use

Overall, the highest prevalence of current e-cigarette use was reported in Indonesia (11.8%) [37], while the lowest prevalence was reported in Thailand (3.3%) [31]. Such differences may be attributable to the implementation of different e-cigarette regulations in these countries. For instance, in Thailand, the importation of e-cigarettes was banned in 2014, followed by the prohibition of sale and services of e-cigarettes in 2015 [39]. On the other hand, not only is Indonesia the only country in Asia that has not ratified the FCTC, sale of e-cigarettes is also not banned [40]. In Malaysia, the regulation of e-cigarettes is under the state jurisdiction and only four states (Johor, Kelantan, Terengganu, Pahang) have banned the sale of e-cigarettes [40]. Additionally, only e-cigarettes containing nicotine are regulated under the Poison Act 1952, while those without nicotine can be sold without any restriction [39,41]. 

Globally, only 29 countries have imposed complete bans on e-cigarette sales, while 45 countries, including Canada, the United States, the countries of the European Union, and the United Kingdom, have used several approaches to regulate the sale of e-cigarettes [42]. Regulations have the potential to affect the use of e-cigarettes. A study involving 14 countries reported that the prevalence of e-cigarette use was lower in countries with strict enforcement of e-cigarette regulations [43]. In these instances, regulations play an important role to help combat the e-cigarette epidemic in adolescents.

### 4.2. Associated Factors of E-Cigarette Use

Sociodemographic factors, including gender, age, ethnicity, school location, academic performance, and sexual experience, play a role in e-cigarette use among adolescents. Several studies consistently reported that male adolescents had higher odds of using e-cigarettes. Similar findings have been reported in the United States [44]. This finding may be explainable as male adolescents are more likely to perceive e-cigarettes as less harmful than smoking cigarettes compared with females [45]. In addition, they also had higher exposure to e-cigarette advertisements on the web, which are accompanied by unproven claims regarding the health benefits of e-cigarette use [46]. The positive association between age and e-cigarette use has been shown in the literature. A study conducted in the United States reported similar findings [47]. A study has reported that peer relationships become more important for males during their mid-adolescence. Hence, they might start using e-cigarettes during this period in order to achieve social acceptance [48]. 

The higher odds of e-cigarette use among Malays and the indigenous people of Sabah and Sarawak have been reported in a study conducted among Malaysian adolescents. This may be due to the higher prevalence of e-cigarette use among adults who are Malays and indigenous people of Sabah and Sarawak compared with adults of Indian and Chinese ethnicity [49]. The same pattern may be observed in adolescents as they tend to imitate their parents’ smoking behaviors [50]. Studies conducted in Thailand and Indonesia did not examine the relationship between ethnicity and e-cigarette use, which may be due to the differences in ethnic composition. For example, in Thailand, almost the entire population is of Thai ethnicity [51]. In this scenario, the influence of ethnicity on e-cigarette use may not be relevant to them and therefore is not being studied.

The higher odds of e-cigarette use among those schooling in urban areas have been noted. This finding may be attributable to the easier access to the internet as websites serve as a platform for information and sales of e-cigarettes [52]. Apart from that, the vape stores are more concentrated in urban than rural areas, and therefore, those schooling in urban areas have easy access to e-cigarettes [36]. In this instance, the use of zoning to restrict the density and location of vape stores may be effective in reducing the availability of and access to e-cigarettes. Such measures have been widely used in alcohol and fast-food retail outlets and have been proven to be successful in reducing alcohol availability and its negative effects on community health [53].

The positive association between poor academic performance and e-cigarette use is in line with another study conducted in Finland [54]. This finding suggests that there are differences in the magnitudes of risk for e-cigarette use in categories of school performance. This could be due to the higher likelihood of skipping class and being alienated from school or studies among e-cigarette users compared to non-users [55]. Sexual experience has also been found to be associated with e-cigarette use. Similarly, a study conducted among adolescents in the Republic of Korea found that those who had sexual experience were more likely to use e-cigarettes [56]. Such associations may be explained by the tendency of exhibiting risky behaviors among e-cigarette users caused by their high levels of sensation seeking as well as low parental support and monitoring [57]. 

There is also evidence to support the relationship between adverse childhood experience with the use of e-cigarettes. Similarly, studies conducted in the United States reported that adverse childhood experiences were associated smokeless tobacco and e-cigarette use [58,59], while another study conducted in Australia reported associations between adverse childhood experiences and e-cigarette use [60]. This may be possibly due to the fact that nicotine is effective in regulating mood among those who experience adverse childhood experiences. Moreover, nicotine may be especially craved during times of chronic distress attributable to adverse childhood experiences [61]. This finding implies that evaluation of adverse childhood experiences and addressing these issues is an important component that should be included in e-cigarette prevention interventions.

The relationship between peer tobacco product use (cigarettes and e-cigarettes) with e-cigarette use in adolescents has also been observed in another study conducted in Republic of Korea [56]. These findings might be expected as adolescents tend to select friends whose participation in risky behaviors is similar to their own [62]. Additionally, adolescents tend to spend more time with their peers, while spending unsupervised time with peers can increase their likelihood of developing problem behaviors and substance use [48]. Similar to our findings, previous studies conducted in Greece [63] and Europe [64] found strong associations between parental tobacco product use with the use of e-cigarettes among adolescents. This finding is explainable as it is common for adolescents to imitate their parents’ smoking behaviors [50]. In addition, it might be difficult for parents who smoke to stop their children from using e-cigarettes due to the inconsistency between their messages and behaviors. 

This review also found that adolescents who had separated parents have higher odds of e-cigarette use. This finding is in line with another study conducted in southern New England which revealed that adolescents with divorced parents were more likely to use e-cigarettes than those with married parents [65]. On the other hand, another study in Finland reported that having an intact family is protective against e-cigarette use in adolescents [54]. These findings are in accordance with the bioecological model of development and the “parental absence perspective”, which suggests that contextual factors, such as the absence of a parent can influence the well-being of an individual [65]. In view of the increased risk for e-cigarette use among adolescents with non-intact families, involving them in support groups for children adjusting to parental divorce may potentially address this issue [66].

Similar to another study conducted in Hong Kong [67], this review found that poor knowledge regarding the harmful effects of e-cigarettes is associated with e-cigarette use in adolescents. Inadequate knowledge of e-cigarettes in adolescents may be due to the lack of formal school-based education about e-cigarettes [68]. School-based education on e-cigarettes has been found to be effective in increasing knowledge about e-cigarettes and reducing the intent to try e-cigarettes among adolescents [69]. Therefore, future school education programs should be enhanced to improve students’ knowledge about the harmful effects of e-cigarettes. 

This review also found that having the perception that e-cigarettes can help to quit smoking and the perception that e-cigarettes are less addictive and do not cause cancer are associated with e-cigarette use. Our findings are in line with other studies which found that adolescents use e-cigarettes because they think that e-cigarettes are safer than conventional cigarettes, while some used e-cigarettes to quit smoking [32,70]. Such findings show that perception regarding e-cigarettes may have great influence in e-cigarette use among adolescents. Nevertheless, scant published data exist with regard to assessing the perception of e-cigarettes among adolescents [71]. Therefore, more studies on perception on e-cigarettes and its relationship with e-cigarette behaviors among adolescents is warranted.

The perception that e-cigarettes could support people quitting cigarette smoking may be related to the campaign from the e-cigarette industry [72] even though the current evidence regarding effectiveness of e-cigarettes for smoking cessation is still inconclusive [73]. This implies that adolescents are susceptible to promotional activities, and therefore it is essential to monitor and control the marketing and advertising activities of e-cigarettes. Apart from that, the wrong perception that e-cigarettes are less addictive and do not cause cancer [73,74] also highlighted the need for more education and awareness about the harm of e-cigarettes for adolescents.

We observed a reported association between current cigarette smoking and e-cigarette use in adolescents, which is consistent with the findings of other studies conducted in Hong Kong [67] and the United States [44]. In view of the high prevalence of dual use of e-cigarettes and conventional cigarettes in adolescents, it may be possible that they were using e-cigarettes as an aid to quit smoking [75]. Furthermore, adolescents who used other tobacco products, including cigarettes, were more likely to perceive e-cigarettes as comparatively safer and therefore more likely to use e-cigarettes [45].

The association between alcohol drinking and e-cigarette use is consistent with findings in Republic of Korea [56] and Hong Kong [67]. This finding is in line with expectation, as a previous study has proven that alcohol, tobacco,, and drug use tend to cluster in adolescents [76]. The clustering of these risky behaviors in adolescents suggests the urgent need for targeted multicomponent health behavior interventions which simultaneously address these risky behaviors. 

In line with another study conducted in the United States [77], we found that accessibility of e-cigarettes plays an important role in e-cigarette use among adolescents. E-cigarettes are easily accessible through online marketplace, social media, and vape shops [36]. A national survey conducted in Malaysia found that more than half (53.2%) of adolescents below the age of 18 years were not prevented from buying e-cigarettes, and most of them obtained their e-cigarettes from friends or retailers [78]. In view of easy access to e-cigarettes among adolescents, enforcement of regulatory interventions to address adolescent access should be considered, for instance, to require e-cigarette retailers to obtain a license for selling these products. Retailer license requirements are an evidence-based method that can be used to reduce youth access to tobacco products as well as their usage in adolescents [79]. 

The positive association between having enough money to buy e-cigarette and the use of cigarettes was observed in a study conducted in Indonesia. In contrast, other studies conducted in Greece [63] and Malaysia [32] did not find such associations. The plausible reason is that some e-cigarettes are cheaper than conventional cigarettes and thus more affordable for adolescents [80].

As with any research, this systematic review is not without limitations. All of the included studies were of cross-sectional design, and therefore causal relationships cannot be determined. Apart from that, the role of publication bias in this systematic review should be acknowledged as grey literature was not included. In addition, language bias must also be considered as we only included English-language publications. Despite these limitations, to the best of our knowledge, this is the first systematic review synthesizing research evidence on prevalence and associated factors of e-cigarette use among adolescents in Southeast Asia. 

## 5. Conclusions

This systematic review showed a substantial prevalence of e-cigarette use among adolescents in Southeast Asia. E-cigarette use is associated with sociodemographic factors, traumatic childhood experience, peer and parental influence, knowledge and perception of e-cigarettes, substance use, and accessibility of e-cigarettes. These factors should not be treated in isolation and should be addressed through multifaceted interventions that simultaneously target multiple factors. Laws, policies, programs, and interventions must be strengthened and tailored to the specific needs of adolescents at risk of using e-cigarettes.

## Figures and Tables

**Figure 1 ijerph-20-03883-f001:**
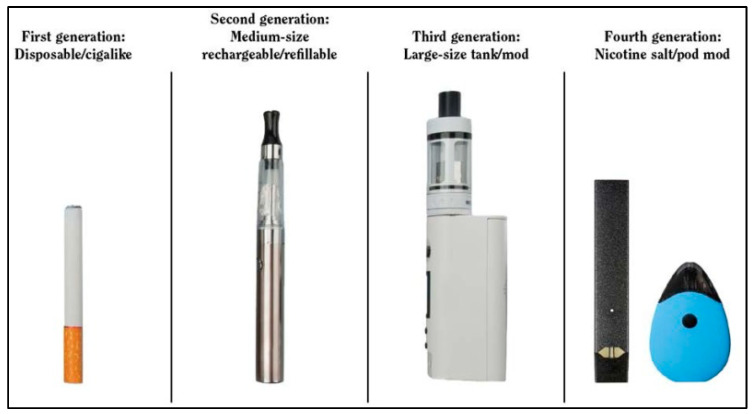
The evolution of e-cigarettes by product generation and characteristics (Source: Chapter 6 Interventions for smoking cessation and treatments for nicotine dependence cessation: a report of the surgeon general) [4].

**Figure 2 ijerph-20-03883-f002:**
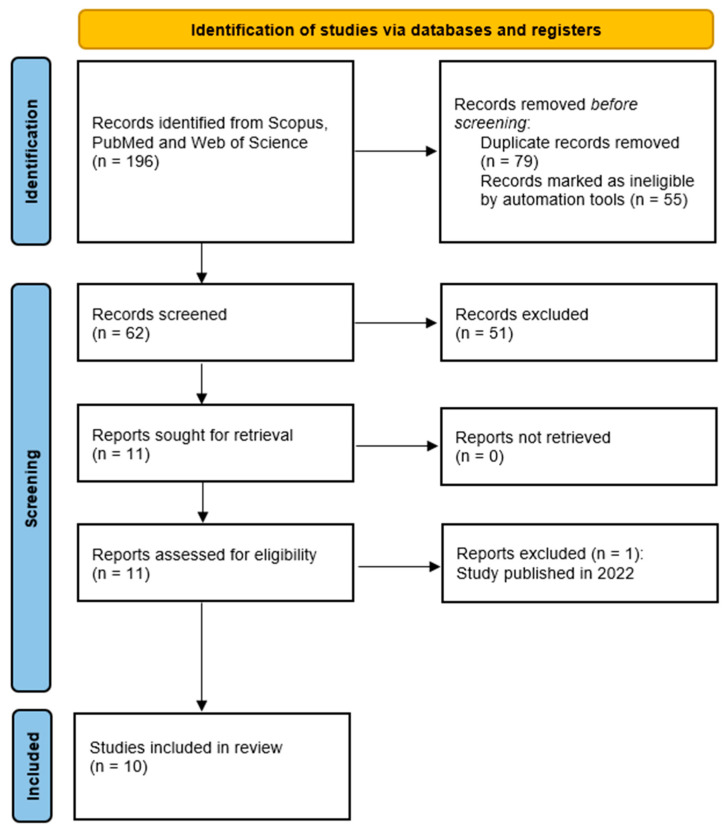
PRISMA flow diagram.

**Table 1 ijerph-20-03883-t001:** Keywords used for searching.

Databases	Search Strings
Scopus	(1) TITLE-ABS-KEY((“prevalence*” OR “use” OR “epidemiology”) AND (“E-cigarette*” OR “vape*” OR “electronic cigarette” OR “Electronic Nicotine Delivery System”) AND (“youth” OR “adolescent*” OR “child”) AND (“Southeast asia” OR “Brunei” OR “Myanmar” OR “Cambodia” OR “Timor-leste” OR “Indonesia” OR “Laos” OR “Malaysia” OR “Philippines” OR “Singapore” OR “Thailand” OR “Vietnam”)) (2) TITLE-ABS-KEY((“risk factor*” OR “predictor” OR “associated factor*” OR “association” OR “correlate*” OR “determinant*”) AND (“E-cigarette*” OR “vape*” OR “electronic cigarette” OR “Electronic Nicotine Delivery System”) AND (“youth” OR “adolescent*” OR “child”) AND (“Southeast asia” OR “Brunei” OR “Myanmar” OR “Cambodia” OR “Timor-leste” OR “Indonesia” OR “Laos” OR “Malaysia” OR “Philippines” OR “Singapore” OR “Thailand” OR “Vietnam”))
Web of Science	(1) ((ALL=(prevalence*)) OR ALL=(use)) OR ALL=(epidemiology) AND (((ALL=(E-cigarette*)) OR ALL=(vape*)) OR ALL=(electronic cigarette)) OR ALL=(Electronic Nicotine Delivery System) AND ((ALL=(youth)) OR ALL=(adolescent*)) OR ALL=(child) AND (((((((((((ALL=(Southeast asia)) OR ALL=(Brunei)) OR ALL=(Myanmar)) OR ALL=(Cambodia)) OR ALL=(Timor-leste)) OR ALL=(Indonesia)) OR ALL=(Laos)) OR ALL=(Malaysia)) OR ALL=(Philippines)) OR ALL=(Singapore)) OR ALL=(Thailand)) OR ALL=(Vietnam) (2) ((((ALL=(risk factor*)) OR ALL=(predictor)) OR ALL=(associated factor*)) OR ALL=(association)) OR ALL=(correlate*)AND (((ALL=(E-cigarette*)) OR ALL=(vape*)) OR ALL=(electronic cigarette)) OR ALL=(Electronic Nicotine Delivery System) AND ((ALL=(youth)) OR ALL=(adolescent*)) OR ALL=(child) AND (((((((((((ALL=(Southeast asia)) OR ALL=(Brunei)) OR ALL=(Myanmar)) OR ALL=(Cambodia)) OR ALL=(Timor-leste)) OR ALL=(Indonesia)) OR ALL=(Laos)) OR ALL=(Malaysia)) OR ALL=(Philippines)) OR ALL=(Singapore)) OR ALL=(Thailand)) OR ALL=(Vietnam)
PubMed	(1) (((((“prevalence*”) OR (“use”)) OR (“epidemiology”)) AND ((((“E-cigarette*”) OR (“vape*”)) OR (“electronic cigarette”)) OR (“Electronic Nicotine Delivery System”))) AND (((“youth”) OR (“adolescent*”)) OR (“child”))) AND (((((((((((((“Southeast asia”) OR (“Brunei”)) OR (“Myanmar”)) OR (“Cambodia”)) OR (“Timor-leste”)) OR (“Indonesia”))) OR (“Laos”)) OR (“Malaysia”)) OR (“Philippines”)) OR (“Singapore”)) OR (“Thailand”)) OR (“Vietnam”)) (2) ((((((((“risk factor*”) OR (“predictor”)) OR (“associated factor*”)) OR (“association”)) OR (“correlate*”)) OR (“determinant*”)) AND ((((“E-cigarette*”) OR (“vape*”)) OR (“electronic cigarette”)) OR (“Electronic Nicotine Delivery System”))) AND (((“youth”) OR (“adolescent*”)) OR (“child”))) AND (((((((((((((“Southeast asia”) OR (“Brunei”)) OR (“Myanmar”)) OR (“Cambodia”)) OR (“Timor-leste”)) OR (“Indonesia”))) OR (“Laos”)) OR (“Malaysia”)) OR (“Philippines”)) OR (“Singapore”)) OR (“Thailand”)) OR (“Vietnam”))

The asterisk symbol (*) was used as a wildcard to increase variability of keywords to be identified.

**Table 2 ijerph-20-03883-t002:** Summary of study location and design.

Study Location	Authors
Thailand	(4) Thepthien et al. 2021 [28], Patanavanich et al. 2021 [29], Ofuchi et al. 2020 [30], Chotbenjamaporn et al. 2017 [31]
Malaysia	(3) Robert Lourdes et al. 2019 [32], Nur Atikah et al. 2019 [33], Yusof et al. 2019 [34]
Indonesia	(3) Kristina et al. 2020 [35], Fauzi & Areesantichai 2020 [36], Bigwanto et al. 2019 [37]
**Study Design**	**Authors**
Cross-sectional	(10) Thepthien et al. 2021 [28], Patanavanich et al. 2021 [29], Ofuchi et al. 2020 [30], Chotbenjamaporn et al. 2017 [31], Robert Lourdes et al. 2019 [32], Nur Atikah et al. 2019 [33], Yusof et al. 2019 [34], Kristina et al. 2020 [35], Fauzi & Areesantichai 2020 [36], Bigwanto et al. 2019 [37]

**Table 3 ijerph-20-03883-t003:** Summary of included studies.

No.	Author (Year)	Title	Study Design	Prevalence	Age Group Involved	Factors
1	Thepthien et al. 2021 [28]	An analysis of e-cigarette and polysubstance use patterns of adolescents in Bangkok, Thailand	Cross sectional	Prevalence of past 30-day e-cigarette use of 6.7%	14–17 years (*n* = 6167)	Gender (male) Grade (year 2 vocational student) Academic achievement (below grade A and B) Sexual experience (ever had sex) Persuaded by close friends to use drugs Peer cigarette use
2	Patanavanich et al. 2021 [29]	Use of E-Cigarettes and Associated Factors among Youth in Thailand	Cross sectional	Prevalence of current e-cigarette use (past 30 day use) was 3.7%; Prevalence of ever e-cigarette use of 7.2%	11–16 years (*n* = 6045)	Gender (male) History of cigarette smoking Parental e-cigarette use Peer e-cigarette use Peer approval of smoking Unaware of e-cigarettes’ risk Poor score on the life asset questionnaire about the power of wisdom
3	Ofuchi et al. 2020 [30]	Adverse Childhood Experiences and Prevalence of Cigarette and E-Cigarette Use Among Adolescents in Bangkok, Thailand	Cross sectional	Prevalence of current e-cigarette use (use in the past 30 days) of 6.7%	13–17 years (*n* = 6167)	Adverse childhood experience: emotional abuse Physical abuse Sexual abuse Parental separation Child violence Incarcerated household Adverse childhood experience score
4	Chotbenjamaporn et al. 2017 [31]	Tobacco use among thai students: Results from the 2015 global youth tobacco survey	Cross sectional	Overall, 3.3% students currently used electronic cigarettes.	13–15 years (*n* = 1721)	Not applicable
5	Robert Lourdes et al. 2019 [32]	Factors Associated With E-Cigarette Usage and the Reasons for Initiation Among Malaysian Adolescents	Cross sectional	Prevalence of current e-cigarette use (use in the past 30 days) of 9.1%	10–19 years (*n* = 13162)	Gender (male) Age (16 to 19 years old) Race (Malays & Sabah and Sarawak Bumiputeras) Current cigarette smokers
6	Nur Atikah et al. 2019 [33]	Factors associated with different smoking statuses among Malaysian adolescent smokers: A cross-sectional study	Cross sectional	Not applicable	13–17 years (*n* = 422)	Gender (male) Recreational cigarette smoker
7	Yusof et al. 2019 [34]	Alternative Tobacco Products Use among Late Adolescents in Kelantan, Malaysia	Cross sectional	The prevalence of current e-cigarette use among late adolescents of 5.9%.	18–19 years (*n* = 388)	Gender (male) Cigarette smoking user Peer use of e-cigarette
8	Kristina et al. 2020 [35]	Trend of electronic cigarette use among students in Indonesia	Cross sectional	Prevalence of current e-cigarette use of 10.65%	16–24 years (*n* = 920)	Gender (male) Knowledge about harm of smoking (Poor) Current cigarette user Attitude toward smoking (neutral) Alcohol consumption (experimental)
9	Fauzi & Areesantichai 2020 [36]	Factors associated with electronic cigarettes use among adolescents in Jakarta, Indonesia	Cross sectional	Prevalence rates of current e-cigarette use among females and males of 0.6% and 8.2%; Overall, 6.3% of females and 29% of males reported ever having used electronic cigarettes.	15–19 years (*n* = 1318)	Gender (male) School locations (Urban) Conventional cigarette smoking use Peer use e-cigarette Easy availability of E-Cigarette Perceptions that electronic cigarettes aid conventional cigarette smoking cessation
10	Bigwanto et al. 2019 [37]	Determinants of e-cigarette use among a sample of high school students in Jakarta, Indonesia	Cross sectional	32.2% of students (*n* = 247) had ever used e-cigarettes and 11.8% of students are current e-cigarette users (*n* = 90)	12–17 years (*n* = 767)	Current smoking of conventional cigarettes Perception that e-cigarettes are less addictive than conventional cigarettes Perception that e-cigarettes do not cause cancer Parental acceptance of e-cigarette use Having enough money to buy e-cigarettes

**Table 4 ijerph-20-03883-t004:** JBI critical appraisal checklist.

Author, Year	Was the Sample Frame Appropriate to Address the Target Population?	Were Study Participants Sampled in an Appropriate Way?	Was the Sample Size Adequate?	Were the Study Subjects and the Setting Described in Detail?	Was the Data Analysis Conducted with Sufficient Coverage of the Identified Sample?	Were Valid Methods Used for the Identification of the Condition?	Was the Condition Measured in a Standard, Reliable Way for All Participants?	Was there Appropriate Statistical Analysis?	Was the Response Rate Adequate, and If Not, Was the Low Response Rate Managed Appropriately?
Thepthien et al. 2021 [28]	Yes	Yes	Unclear	Yes	Yes	Yes	Yes	Yes	Yes
Patanavanich et al. 2021 [29]	Yes	Yes	Unclear	Yes	No	Yes	Yes	Yes	Yes
Ofuchi et al. 2020 [30]	Yes	Yes	Unclear	Yes	Yes	Yes	Yes	Yes	Yes
Chotbenjamaporn et al. 2017 [31]	Yes	Yes	Unclear	Yes	No	Unclear	Unclear	Yes	Unclear
Robert Lourdes et al. 2019 [32]	Yes	Yes	Yes	Yes	Yes	Yes	Yes	Yes	Yes
Nur Atikah et al. 2019 [33]	Yes	Yes	Unclear	Yes	Yes	Yes	Yes	Yes	Yes
Yusof et al. 2019 [34]	Yes	Yes	Yes	Yes	Yes	Yes	Yes	Yes	Yes
Kristina et al. 2020 [35]	Yes	Yes	Yes	Yes	Yes	Yes	Yes	Yes	Yes
Fauzi & Areesantichai 2020 [36]	Yes	Yes	Yes	Yes	Yes	Yes	Yes	Yes	Yes
Bigwanto et al. 2019 [37]	Yes	Yes	Yes	Yes	Yes	Yes	Yes	Yes	Yes

## Data Availability

Not applicable.

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
