# Peer review of "Prevalence and Associated Factors of E-Cigarette Use among Adolescents in Southeast Asia: A Systematic Review"

_ijerph, 2023, doi:10.3390/ijerph20053883_

Round 1
Reviewer 1 Report
This is a well written manuscript. Search strategy is elaborately displayed.
Only a few queries:
1. Search strategy mentioned in the eligibility criteria as well as Abstract tells us is from 2012 to 2021, yet one study from 2022 is included in the final analysis; it needs explanation.
2. There should be few sentences or a paragraph about Southeast Asia and the countries included in this area signifying their importance in the background of current study.
3. References list looks overwhelming and sometimes citations are burdened. Please revisit your reference list and use the best ones for citation.
Reviewer 2 Report
In this review, the authors present and analyze the prevalence and factors associated with e-cigarette use among adolescents in Southeast Asia together with future recommendations towards its prevention. In this systematic review of 11 studies, they summarized the prevalence of e-cigarette use among adolescents in Southeast Asia. They identified several associated factors of e-cigarette use, which are classified into six categories: 1) Sociodemographic, 2) Traumatic childhood experience, 3) Peer and parental influence, 4) Knowledge and perception of e-cigarettes, 5) Substance use, and 6) Accessibility of e-cigarettes. They suggested that understanding the factors associated with e-cigarette use in adolescents will serve as a guide to support development and implementation of targeted interventions. They also suggested that evaluation of adverse childhood experiences and addressing these issues is an important component that should be included in e-cigarette prevention interventions. They concluded that "factors 1-6 should not be treated in isolation and should be addressed through multifaceted interventions that simultaneously target multiple factors. Laws, policies, programmes and interventions must be strengthened and tailored to the specific needs of adolescents at risk of using e-cigarettes".
I have some question and comments:
1.- Although there are many articles available on different aspects of e-cigarettes, an overview of the current state of e-cigarette use among adolescents in Southeast Asia is highly desirable. This review seems to fill this need. The review is a collection of data following the Systematic reviews and Meta-Analyses (PRISMA) statement, published in 2009, which was designed to help systematic reviewers to transparently report: i) why the review was conducted, ii) what the authors did and ii) what did they find?. I think it is important for the authors to highlight aspects i) to iii) in the introduction to better understand the current status of e-cigarette use among adolescents in Southeast Asia.
2.- How many reviews have been published about it?.
The work is interesting. I think it is suitable for the journal.
Reviewer 3 Report
The paper entitled * Prevalence and Associated Factors of E-Cigarette Use 2 among Adolescents in Southeast Asia: A Systematic Re- 3 view * study the use of e-cigarettes in adolescents remains a major public health concern. The reporting of this systematic review is in accord- 15 ance with the Preferred Reporting Items for Systematic Reviews and Meta-Analyses 16 (PRISMA) 2020 statement. We carried out a literature search through three databases (Sco- 17 pus, PubMed, Web of Science) and targeted original English-language articles published be- 18 tween 2012 and 2021. A total of 11 studies were included in this review. The prevalence of 19 current e-cigarette uses ranges between 3.3% to 11.8%. Several associated factors of e-ciga- 20 rette use were identified, including sociodemographic factors, traumatic childhood experi- 21 ence, peer and parental influence, knowledge and perception, substance use and accessibility 22 of e-cigarettes. These factors should be addressed though multifaceted interventions which 23 simultaneously target multiple factors. There some points are needed for more explanation then the manuscript will be suitable for publication.
1-Introduction part is short and must be enlarged with more updated references.
2- Materials and Methods part:
- In Study Selection part: Why authors depend on three reviewers only?
- In Critical Appraisal and Data Extraction: Quality appraisal was conducted by using the Mixed Method
Are any others method can be used to confirm the resulted data?
3-In Results part:
-In Prevalence of E-cigarette use part: author's use of 10 articles mentioned about the prevalence of current e-cigarette 131 smoking past 30 days use. I recommended that period to be past 90 days use and make more search during this period.
- In Traumatic Childhood Experience : authors stated that only one study focuses on traumatic childhood experience as a risk factor for 162 e-cigarette use among adolescent I recommended to explain more studies in north America and make comparison with current study in southeast Asia .
Round 2
Reviewer 3 Report
This manuscript deals with an important topic and discuss the significant risk of Electronic cigarettes, Some points must be added:
1-If authors can add images for types of this e-cig with some explanation about it.
2- There is a critical study that authors may benefit from adding the effect of e-cig on oxidative stress and deleterious effect on the lung, it will reinforce the discussion part significantly and may get a recommendation based on the current manuscript,
https://doi.org/10.3390/cryst12070972
